Open Peer Review | Computational Biology | Methods and Protocols

# Unfolding and de-confounding: biologically meaningful causal inference from longitudinal multi-omic networks using METALICA

Daniel Ruiz-Perez,[1] Isabella Gimon,[1] Musfiqur Sazal,[1] Kalai Mathee,[2,3] Giri Narasimhan[1,3]

**ABSTRACT**  A key challenge in the analysis of microbiome data is the integration of multi-omic datasets and the discovery of interactions between microbial taxa, their expressed genes, and the metabolites they consume and/or produce. In an effort to improve the state of the art in inferring biologically meaningful multi-omic interactions, we sought to address some of the most fundamental issues in causal inference from longitudinal multi-omics microbiome data sets. We developed METALICA, a suite of tools and techniques that can infer interactions between microbiome entities. METALICA introduces novel unrolling and de-confounding techniques used to uncover multi-omic entities that are believed to act as confounders for some of the relationships that may be inferred using standard causal inferencing tools. The results lend support to predictions about biological models and processes by which microbial taxa interact with each other in a microbiome. The unrolling process helps identify putative intermediaries (genes and/or metabolites) to explain the interactions between microbes; the de-confounding process identifies putative common causes that may lead to spurious relationships to be inferred. METALICA was applied to the networks inferred by existing causal discovery, and network inference algorithms were applied to a multi-omics data set resulting from a longitudinal study of IBD microbiomes. The most significant unrollings and de-confoundings were manually validated using the existing literature and databases.

**IMPORTANCE**  We have developed a suite of tools and techniques capable of inferring interactions between microbiome entities. METALICA introduces novel techniques called unrolling and de-confounding that are employed to uncover multi-omic entities considered to be confounders for some of the relationships that may be inferred using standard causal inferencing tools. To evaluate our method, we conducted tests on the inflammatory bowel disease (IBD) dataset from the iHMP longitudinal study, which we pre-processed in accordance with our previous work. From this dataset, we generated various subsets, encompassing different combinations of metagenomics, metabolomics, and metatranscriptomics datasets. Using these multi-omics datasets, we demonstrate how the unrolling process aids in the identification of putative intermediaries (genes and/or metabolites) to explain the interactions between microbes. Additionally, the de-confounding process identifies potential common causes that may give rise to spurious relationships to be inferred. The most significant unrollings and de-confoundings were manually validated using the existing literature and databases.

**KEYWORDS**  longitudinal microbiome analysis, multi-omic integration, causal inference, unfolding, de-confounding

Microbiomes are communities of microbes inhabiting an environmental niche. Metagenomics data sets contain sequenced reads from samples of a microbial

Address correspondence to Giri Narasimhan, giri@fiu.edu.

The authors declare no conflict of interest.

See the funding table on p. 14.

community and are used to infer a detailed abundance profile of the microbial taxa present in that community (1, 2). More recently, additional types of biological data are being generated from microbiome studies, including but not limited to (i) metatranscriptomics and metaproteomics, which help survey the expression of the totality of genes and proteins in the microbial community (3); (ii) metabolomics, which helps profile the concentrations of the entire set of small molecules (metabolites) present in the microbiome's environmental niche (4); (iii) metaresistomics, which helps capture the repertoire of antibiotic resistance genes present in the microbial community (5); and (iv) host transcriptomics, which provides information about the expression levels of the host genes (6). Such multi-omic data sets are critical for a more in-depth and functional understanding of microbial communities. They also shed light on some of the interactions between the entities in the microbiome (7). Thus, the study of microbial communities offers a powerful approach for inferring interactions within the community (8, 9), their impact on the host environment (5), and their role in disease and health (10, 11).

A major bioinformatic challenge is the "integrative" analysis of multi-omic data sets from microbiomes (12). Most multi-omic studies focus on a separate analysis of each omic data set without building a unified model (13). There have been some attempts (14–18) to build tools and develop techniques to facilitate an integrative analysis (19, 20). Significant advances were recently made on analyzing multi-omic longitudinal data sets by Ruiz-Perez et al. (21). Questions related to reproducibility, flexibility, interpretation, and biological validity continue to be challenges in the area of multi-omic microbiome analysis (21–23).

Deep learning approaches for integrating multi-omics (24, 25) have also been developed, but they are either hard to interpret or limited to predicting just one of the omic profiles. Additionally, the high computational cost of deep learning further prevents these models from being useful at providing insights into the interplay between the different omic entities. Partial least squares models have also been used to facilitate this integration (26). Their limitations depend on the underlying data generation model and are generally prone to produce spurious results when applied to high-dimensional data sets (27).

Given that microbiomes are inherently dynamic, longitudinal multi-omic data sets are important to fully understand the complex interactions that take place within these communities (28). Many attempts have been made to analyze data from longitudinal studies (17, 18, 29); however, these approaches do not attempt to study interactions between taxa. An alternative approach involves the use of dynamical systems such as the generalized Lotka-Volterra (gLV) models (30, 31). As was noted by Ruiz-Perez et al. (21), the large set of parameters in these probabilistic models diminishes their utility for use in inference.

In previous work (21, 32), we have described sophisticated methods to model and analyze data from longitudinal microbiome studies using dynamic Bayesian networks (DBNs). Our approach involved starting from next-generation sequencing data and other omics measurements. Every attempt was made to ensure that the resulting networks had biologically meaningful edges and were not a result of overfitting. However, even if an edge was directed from an entity measured at a previous time point to an entity measured at a later one, it did not guarantee that it represented a true and direct causal interaction. It could be possible for the edge to be merely the result of a statistical correlation caused by an indirect causal relationship or model overfitting.

Microbiomes are complex environments with many subtle relationships. However, causal discovery relies on noisy data from error-prone technologies and has to contend with a host of hidden confounders that may be hard or impossible to identify, let alone be measured. The jump to infer causality is a natural next step in understanding multi-omic interactions, and the lack of research in this area is striking. Most of the causal microbiome literature focuses on the causal impact of the microbiome on health or disease, but not on the causal interactions between these microorganisms (33–36).

This shortcoming was addressed in our previous work (10, 11). Finally, another major challenge in building true models of biological interactions lies in developing methods to validate them and in providing confidence measures.

## MATERIALS AND METHODS

### Overview

In this section, we have considered three network learning methods, dynamic Bayesian networks (DBNs) using PALM (21), TETRAD (37–39), and Tigramite (40), and applied them to a rich, multi-omics data set. We then describe unrolling, a novel method to extract well-supported, biologically relevant conjectures on entities that appear to mediate complex relationships between microbes in a microbiome. Finally, we describe de-confounding, another novel method to identify network edges for which there is strong support for conjecturing that they are spurious, i.e., not causal. The two methods, unrolling and de-confounding, constitute the heart of the METALICA (MicrobiomE Temporal AnaLysIs using CAusality) package presented here.

In what follows, we describe the experiments that were performed. We start by describing the data sets used for the experiments and the preprocessing of the data. Next, we discuss the theory behind the first of the network learning methods, i.e., DBNs, and follow it up with the constraining structures used and the procedure to create a collection of DBNs with the help of PALM. This is followed by a brief description of two well-known methods, TETRAD and Tigramite, to create causal networks for the above data set. Finally, we describe the methods of unrolling and de-confounding to evaluate and compare the causal discoveries made by all the three network learning algorithms.

### Data sets

To test the three proposed methods, the inflammatory bowel disease (IBD) cohort from a study that included 132 individuals across five clinical centers was used (18). During a period of 1 year, each subject was profiled (biopsies, blood draws, and stool samples) every 2 weeks on average. This yielded temporal profiles for the metagenomes, metatranscriptomes, metaproteomes, metabolomes, and viromes across all subjects. Additionally, for each subject, host- and microbe-targeted human RNA sequencing was yielded from biopsies collected at the initial screening colonoscopy sampled from two sites in the gut (ileum and rectum) to obtain the host transcriptomic profile. All data are fully described and available at https://ibdmdb.org.

### Preprocessing the data

We used the processed version of the IBD data set generated by our previous work (21), which provided temporally aligned and unaligned versions of metagenomics, metatranscriptomics, metabolomics, and host transcriptomics data. As explained in Ruiz-Perez et al. (21), the data were normalized and centered, the time series were smoothed, and then temporally aligned. For completeness, a summary of this process is described here. The different omics data types were processed separately. First, the taxon, metabolite, and gene abundance values were normalized to make each type separately add up to 1 for each subject, thus expressing each abundance value as a fraction of the whole metagenome, metabolome, and metatranscriptome. Then, the intensities of the metabolites and genes were scaled to match the mean of the taxa because the larger number of genes and metabolites had made their average values much smaller. Metabolites without an HMDB ID or with near-zero variance over the originally sampled time points were removed. Any sample that had less than five measured time points in any of the multi-omics measurements was also removed. The multi-omic time series were then smoothed using B-splines to deal with irregular sampling rates and missing time points. Then, temporal alignment of the time series data from individuals was performed as described in Lugo-Martinez et al. (32). This was done because they assumed that even though the underlying biological process of the different subjects may be the same, the

speed at which the processes occur in each patient could be different. These temporal alignments use a linear time transformation function to "warp" one-time series into a common, representative sample time series used as the "reference" (32), which was selected as follows for each omics data. All possible pairwise alignments were generated between them and the time series that resulted in the least total overall error in the alignments was selected as the reference. Abnormal and noisy samples from the resulting set of alignments were filtered out. Given an individual's warped/aligned time series for a specific omic type (represented by a transformation), the other multi-omics data were also aligned using the same transformation. The resulting data set comprises 51 sets of multi-omics time series, 1 set per subject. We also further restricted ourselves to just the Crohn's disease patients for some analyses, which after the same filtering as described above, resulted in 11 patients.

Due to the relatively small number of time points in each time series, new data sets were generated by simply increasing the sampling frequency from each smoothed time series. Thus, a time series with a sampling rate of 7 days was created. The three preprocessed omics data were then separated, resulting in sets denoted by $\mathbb{T}, \mathbb{G}$, and $\mathbb{M}$, representing the data involving just taxa, genes, and metabolites, respectively. They were also combined to generate different subsets and denoted in a natural way by concatenating the individual symbols. The resulting data sets were the temporally aligned and unaligned versions of the following: $\{\mathbb{T}, \mathbb{G}, \mathbb{M}, \mathbb{TG}, \mathbb{TM}, \mathbb{GM}, \mathbb{TGM}\}$.

In an effort to increase the number of biologically interpretable results and to get the most significant validations of the interactions, the attributes that were cataloged in KEGG (41) were used. This resulted in the selection of 27 bacterial species, 34 genes, and 19 metabolites in addition to 1 so-called "clinical" variable (sampling time, represented by the week during which the sample was obtained). The process described above is generalizable, meaning that more omics data sets, metadata, and clinical variables can be added with relative ease.

## Dynamic Bayesian networks

DBNs are a variety of Bayesian networks (BNs) designed to represent temporal connections between variables as their edges represent lagged dependencies. DBNs can be used to conduct time-varying probabilistic inference and causal discovery. They were developed to unify models such as Kalman filters, autoregressive-moving-average models (ARIMA), and hidden Markov models (HMMs) into a general probabilistic model and inference mechanism (42, 43) and are conceptually similar to probabilistic Boolean networks (PBN) (44). DBNs can model the types of relationships supported by the above methods and can capture even more complex relationships with both discrete and continuous variables conditioned on either temporal or non-temporal variables.

This work focuses on a version of DBNs called two-timeslice BN (2TBN) (45), which finds relationships between variables over adjacent time steps. Let $X_i^t$ denotes the value of variable $X_i$ at time $t$. It can be calculated from the internal regressors if the values of the other variables are known at the previous time point, $t - 1$. We employed a tool called PALM, which uses a multi-omics DBN model proposed by Ruiz-Perez et al. (21). PALM integrates different omics data sets with flexible structure constraints. In particular, we also used their proposed Skeleton and Augmented constraints. These constraints are described below in "Constraining structures." Idealized DBN construction methods require an exponential-time exhaustive search using all subsets of nodes. However, it is possible to construct DBNs more efficiently by limiting the number of "parents" for each node (i.e., bounding the number of incoming edges for each node).

## Constraining structures

The above input was fed into PALM (21). The set of allowable edges was constrained by providing a Skeleton structure as input to the DBN construction step as described by Ruiz-Perez et al. (21). These constraints, which are provided in the form of a matrix,

only allow edges between certain types of nodes, greatly reducing the complexity of searching over possible structures and preventing over-fitting. Specifically, intra edges (i.e., edges within the same time point) from taxa nodes to gene (expression) nodes and from gene nodes to metabolites (concentration) nodes were allowed. All other interactions within the same time point (e.g., direct gene to taxa) were disallowed. In addition, inter edges (i.e., edges between nodes from adjacent time points) were only allowed from metabolites to taxa nodes in the next time point, and self-loops, i.e., edges from node $X_i^t$ to $X_i^{t+1}$ for all types of nodes. (Note that, whenever it is obvious by the context, random variables and the nodes in the networks that represent them are not differentiated.) The restrictions in the Skeleton reflect the basic ways the different entities interact with each other, i.e., taxa express genes that they carry on their genomes; these, in turn, are involved in metabolic pathways for the synthesis of metabolites; subsequently, the metabolites impact the growth of taxa (in the next time slice).

A less constrained framework referred to as the Augmented skeleton was also used to produce an alternative set of networks. Unlike the original Skeleton, the Augmented framework also allows intra edges from taxa to metabolites to account for cases where noise or other issues related to gene-profiling may limit our ability to indirectly connect taxa and the metabolites they produce. All other edges from the skeleton were retained.

## Computing DBNs using PALM

DBNs were learned using PALM for all subsets of the omics data sets from "Data sets" above (i.e., $\{\mathbb{T}, \mathbb{G}, \mathbb{M}, \mathbb{TG}, \mathbb{TM}, \mathbb{GM}, \mathbb{TGM}\}$), for several different numbers of allowable parents ($\{3, 4, 5, 6\}$), for temporally aligned and unaligned data sets, and for the skeleton and augmented constraint frameworks, thus resulting in a total of $7 \times 4 \times 2 \times 2 = 112$ potential DBN networks. A total of 100 networks were learned by subsampling subjects with replacement (i.e., 100 bootstrap repetitions) for each model. The networks were then combined, averaging the regression coefficient (weight) of the edges as long as they appeared in at least 10% of the repetitions. Each edge was also labeled with the bootstrap score or support (proportion of times that edge appears). Each repetition was set to run independently on a separate processor using Matlab's Parallel Computing Toolbox.

In order to explore causal inferencing, two other well-known methods (TETRAD and Tigramite) (37–40) were applied on our data sets. Note that the exact same set of nodes was used as those in the two-time-slice DBN, meaning that every microbiome quantity (taxon abundance, gene expression, metabolite concentration) is represented by two nodes, one from a "previous" time instant and one from the "current" time instant. Since all the networks were on the same set of nodes, it facilitates the comparison between all three methods. We also note that TETRAD and Tigramite do not learn based on a global score such as likelihood, but rather on conditional independence tests.

## Causal networks using the TETRAD suite

The tsGFCI (SVAR-GFCI) (46) algorithm is implemented in the TETRAD package (37–39), for which the wrapper PyCausal (47) was used. The tsGFCI algorithm is a version of tsFCI (48) and GFCI, while tsFCI is, in turn, the evolution of FCI (49). FCI is, in turn, a modification of PC-stable, which was designed by modifying PC, an adaptation of the SGS algorithm (50).

The algorithm tsFCI (SVAR-FCI) is based on a modified version of the FCI algorithm. Briefly, it uses the direction of time to orient interactions and enforces repeating structures for both adjacencies and orientations based on the stationarity assumption. Since the hybrid score-based GFCI is usually more accurate in finite samples than FCI, similar modifications were made in the development of tsGFCI. In this case, a greedy initial adjacency search is used, enforcing time order and repeating structures, and scores the structures using BIC (51).

For each significance threshold $\alpha \in \{0.0001, 0.001, 0.01, 0.1\}$, different networks were learned with the PositiveCorr CI test, the FisherZScore network score, and for each

combination of omics data sets and alignment. A total of $4 \times 7 \times 2 = 56$ experiments were performed with TETRAD. Each TETRAD experiment was repeated with $N$ bootstrapping repetitions. Here, $N = 10$ was used.

## Causal networks with Tigramite

For the discussion below, the following notation is needed. Let $\mathrm{Pa}_G(X)$ represents the parents of node $X$ in network $G$. When the context is clear, $G$ is dropped and simply denoted as $\mathrm{Pa}(X)$. Let $\mathrm{Pa}^p(X)$ denotes the $p$ "strongest" parents. Independence of $A$ and $B$ conditioned on $C$ is denoted by $A \perp\!\!\!\perp B|C$. Tigramite (40) implements the PCMCI algorithm, which works in two stages—conditional selections followed by causal discovery.

### Conditional selections

A modified version of the PC-stable algorithm (adapted for time series and with the skeleton constraints) is used to compute a set of variables that are inferred to have a causal effect on each node $X$. It obtains the set of parents, $\mathrm{Pa}_G(X_i)$, estimated from the data (which may be superset of the true set) for all variables $X_i, i = 1, ..., n$. This is achieved as follows. For every variable, the set of parents are initialized to all allowable parents. Then, conditional independence tests are applied for each edge, $(X_i^{t'}, X_j^t)$, using conditioning sets of increasing size, removing the edge as soon as a test fails. (Note that, as per our constraints, $t' = t$ or $t' = t - 1$.) In each case, the null hypothesis states that the two variables at the endpoint of the edge being considered remain dependent even when conditioned on an appropriate set of size $p \geq 0$, as stated below:

$$H_0 : X_i^{t'} \not\perp\!\!\!\perp X_j^t \mid S, \text{ for any } S \subseteq \mathrm{Pa}(X_j^t) \setminus \{X_i^{t'}\}, \text{ with } |S| = p. \tag{1}$$

The rejection of the null hypothesis $H_0$ requires a significance threshold $a$. All possible sets $S \subseteq \mathrm{Pa}(X_j^t) \setminus \{X_i^{t'}\}$ with cardinality $p$ are considered such that $1 \leq p \leq q_{\max}$.

### Causal discovery stage

Next the MCI algorithm is applied, which employs a more stringent conditional independence test, for each surviving edge $X_i^{t'} \rightarrow X_j^t$, retaining it if and only if

$$X_i^{t'} \not\perp\!\!\!\perp X_j^t \mid \left(\mathrm{Pa}(X_j^t) \setminus \{X_i^{t'}\}\right) \cup \mathrm{Pa}^p(X_j^t). \tag{2}$$

Since Tigramite assumes that all the data points belong to a single subject, bootstrap cannot be implemented in the usual way of subsampling subjects with replacement. Instead, a different network was learned for each subject, and the resulting networks were then combined. The percentage of times that a given edge appears in all the different networks was annotated in the edge, together with the averaged cross-link strength. Different networks were learned for different significance threshold values, $\alpha \in \{0.0001, 0.001, 0.01, 0.1\}$, for each CI test available (GPDC, CMIknn, ParCorr) (40), and for each omics data set. A total of $4 \times 3 \times 7 \times 2 = 168$ experiments were performed with Tigramite.

The following sections introduce the two causal network analysis techniques in METALICA, which will be applied to the networks learned with the methods introduced in "Computing DBNs using PALM," "Causal networks using the TETRAD suite," and "Causal networks with Tigramite" using DBNs, TETRAD, and Tigramite.

## Unrolling

Typical algorithms for network learning and analysis fail to elucidate the actual reasons why two entities may be causally related to each other. An important challenge in microbiome analysis is to use multi-omics data to determine whether and how two

taxa may be interacting with each other. The term "unrolling" is hereby introduced as the process of determining the sequential steps by which two omic entities potentially interact with each other. This is done by learning independent networks using different subsets of omics data. For example, by learning two separate networks with the $\mathbb{T}$ and the $\mathbb{TM}$ data sets, an interaction between two microbial taxa (as suggested by the former) can be surmised to be via metabolic intermediaries (as suggested by the latter).

To make this more formal, let $G_{\mathbb{X}} = (V_{\mathbb{X}}, E_{\mathbb{X}})$ represents the network learned using data set $\mathbb{X}$, with vertex set $V_{\mathbb{X}}$ and edge set $E_{\mathbb{X}}$. Now, an explanation by unrolling occurs if the following three conditions are true:

i. There is an edge from $T_i$ to $T_j$ in $G_{\mathbb{T}}$, for some $T_i, T_j \in V_{\mathbb{T}}, i \neq j$.
ii. There is no edge from $T_i$ to $T_j$ in the network $G_{\mathbb{TM}}$.
iii. There exists some metabolite $M_x \in V_{\mathbb{TM}}$ such that edges $(T_i, M_x)$ and $(M_x, T_j)$ exist in $G_{\mathbb{TM}}$.

If the above three conditions are met, the interaction between the taxa $T_i$ and $T_j$ is inferred to be happening through an intermediary metabolite $M_x$, which is "produced" by $T_i$ and "consumed" by $T_j$.

This process can be replicated by unrolling the edges of the network inferred from $\mathbb{T}$ with the one inferred from $\mathbb{TG}$ to discover the genes that are likely driving the interaction between the same pair of taxa. Finally, the networks $G_{\mathbb{TG}}$ from $\mathbb{TG}$ or $G_{\mathbb{TM}}$ from $\mathbb{TM}$ can be unrolled using the more detailed network $G_{\mathbb{TGM}}$ to find fully unrolled chains of the form $T_i \rightarrow G_y \rightarrow M_x \rightarrow T_j$ in $G_{\mathbb{TGM}}$ with the capability to simultaneously explain the edges $T_i \rightarrow T_j$ in $G_{\mathbb{T}}$, the chain $T_i \rightarrow M_x \rightarrow T_j$ in $G_{\mathbb{TM}}$, and the chain $T_i \rightarrow G_y \rightarrow T_j$ in $G_{\mathbb{TG}}$.

This step-wise unrolling is necessary to discover relationships with strong support from the data, where the network learned from $\mathbb{T}$ was unrolled in a network learned from some subset of $\{\mathbb{TG}, \mathbb{TM}, \mathbb{TGM}\}$. The number of the networks from $\{\mathbb{TG}, \mathbb{TM}, \mathbb{TGM}\}$ that support the unrolling provides a degree of confidence for that unrolling. Furthermore, the bootstrap score for each of the edges involved in the process is reported, together with an overall score that is computed as the product of the individual bootstrap scores of the two replacement edges. This unrolling approach is explained with concrete examples in Discussion under "Uncovering unrolled biological relationships."

## De-confounding

Most current causal inference techniques rely on the causal sufficiency assumption, which assumes that there are no hidden confounders (for any pair of variables) in the data. Confounders are variables that are (i) unknown, (ii) known but not measured, or (iii) measured but not used in the analysis but affect both the cause and the effect of at least one predicted interaction. Predictions of interactions with hidden confounders could be incorrect. The strength of a predicted interaction may be enhanced or diminished when the hidden confounder is not used in the analysis. It is also possible that the predicted interaction may introduce spurious edges when the hidden confounder is not used in the analysis.

In general, the causal sufficiency assumption may be "too strong" and may be impossible to verify, even with the availability of richer data sets that include multi-omics data, thus making this assumption a key obstacle to performing accurate causal inference (52). Going beyond the multi-omic domain, causal sufficiency is an assumption that does not strictly hold in most observational data sets since it is difficult or impossible to include all possible explanatory variables in a study.

A recent paper by Wang and Blei (53) attempts to perform de-confounding, which is the process of removing the effect of all confounders. They introduce the concept of "substitute confounders," which attempts to account for the effect of all hidden confounders in order to arrive at unbiased estimates of causal effects. A major limitation of their method is that the de-confounded interactions are not identified, which is

important for understanding the interactions. Furthermore, there may not be a one-to-one correspondence between the substitute confounder and some real confounder, meaning that one substitute confounder may be an approximation for a combination of several hidden confounders.

In this work, a different approach for the task of de-confounding interactions is taken, inspired by the unrolling approach described above. Independent networks are iteratively learned with different subsets of data with the hope that by adding a new omics layer it would be possible to identify some of the relevant intermediate entities and the corresponding interactions. As before, $G_{\mathbb{X}} = (V_{\mathbb{X}}, E_{\mathbb{X}})$ represents the network learned using data set $\mathbb{X}$, with vertex set $V_{\mathbb{X}}$ and edge set $E_{\mathbb{X}}$. For example, by learning a network with the $\mathbb{T}$ and $\mathbb{TM}$ data sets, interactions can be de-confounded if the following three conditions are satisfied:

i. There is an edge $(T_i, T_j)$ in $G_{\mathbb{T}}$, i.e., $(T_i, T_j) \in E_{\mathbb{T}}$, for some $T_i, T_j \in V_{\mathbb{T}}, i \neq j$.

ii. There is *no* edge from $T_i$ to $T_j$ in $G_{\mathbb{TM}}$, i.e., $(T_i, T_j) \notin E_{\mathbb{TM}}, i \neq j$.

iii. Edges $(M_x, T_i)$ and $(M_x, T_j)$ exist in $G_{\mathbb{TM}}$, i.e., $(M_x, T_i), (M_x, T_j) \in E_{\mathbb{TM}}, i \neq j$, for some metabolite $M_x \in V_{\mathbb{TM}}$.

Using this method, if the above conditions are satisfied for a pair of taxa, $T_i$ and $T_j$, the direction for the directed edge $(T_i, T_j) \in E_{\mathbb{T}}$ is deduced and the inferred interaction between the two taxa is spuriously introduced by the metabolite $M_x$ acting as a confounder. The metabolite can also be inferred to impact the abundance of both taxa, $T_i$ and $T_j$. One possible scenario is that the metabolite, $M_x$, could be an essential metabolite for both taxa, and its presence or absence from the data could make the abundance of the taxa to appear correlated.

As with metabolites, this process can be repeated by de-confounding $G_{\mathbb{T}}$ with edges from $G_{\mathbb{TG}}$ to discover genes/proteins that could confound a presumed causal connection between the taxa. In general, the networks learned using the $\mathbb{T}, \mathbb{G}$, and/or $\mathbb{M}$ data sets can be de-confounded by the networks learned using one or more of the data sets from $\{\mathbb{TG}, \mathbb{TM}, \mathbb{GM}, \mathbb{TGM}\}$. Similarly, networks learned using one of $\mathbb{TG}, \mathbb{TM}$, or $\mathbb{GM}$ data sets can be de-confounded by the networks learned using $\mathbb{TGM}$. This could lead to chains of de-confoundings, where an interaction that led to the de-confounding a relationship is itself later de-confounded.

As before, for each de-confounding discovery, the following is reported: (i) the confounded edge, (ii) the de-confounder, (iii) the bootstrap score for the edges involved in the discovery, (iv) the overall score of the discovery computed as the product of the individual bootstrap scores of the two replacement edges, and (v) the two data sets that were used to discover the specific de-confounding. The results of the de-confounding approach are explained with examples in Discussion.

## RESULTS

A large number of networks were learned with the different data subsets, the different methods, and the parameter settings, as mentioned above in "Computing DBNs using PALM," "Causal networks using the TETRAD suite," and "Causal networks with Tigramite," respectively, for DBN, TETRAD, and Tigramite. Unrolling and de-confounding were implemented in METALICA and applied to all the resulting networks, as described in Materials and Methods. The results from the experiments are presented below.

### Resulting networks

Figure 1 shows the DBNs learned from the $\mathbb{T}, \mathbb{TM}, \mathbb{TG}$, and $\mathbb{TGM}$ versions of the Crohn's disease data sets without temporal alignment. The structure of the networks learned by the other tools was similar to those shown and can be found in the supplemental material. Self loops were hidden in the visualization to avoid unnecessary clutter. The

## a) Taxa

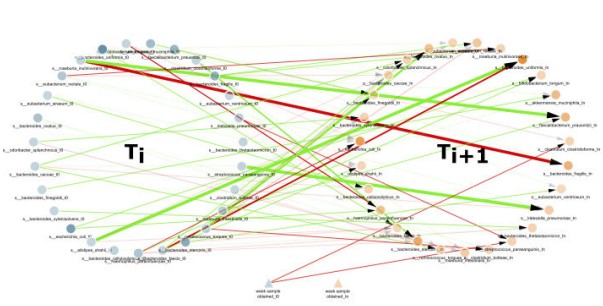

## b)Taxa & Metabolites

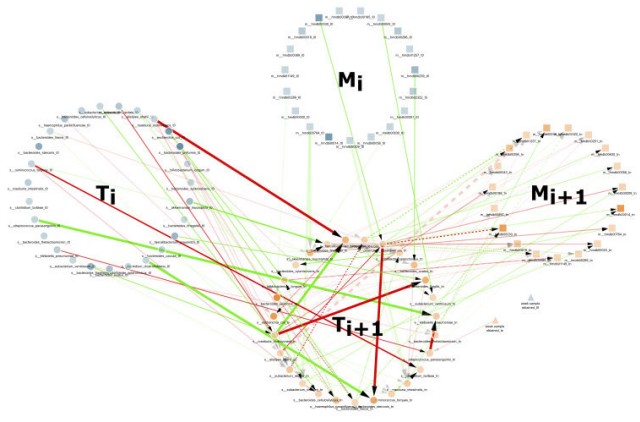

## c) Taxa & Genes

## d) Taxa & Genes & Metabolites

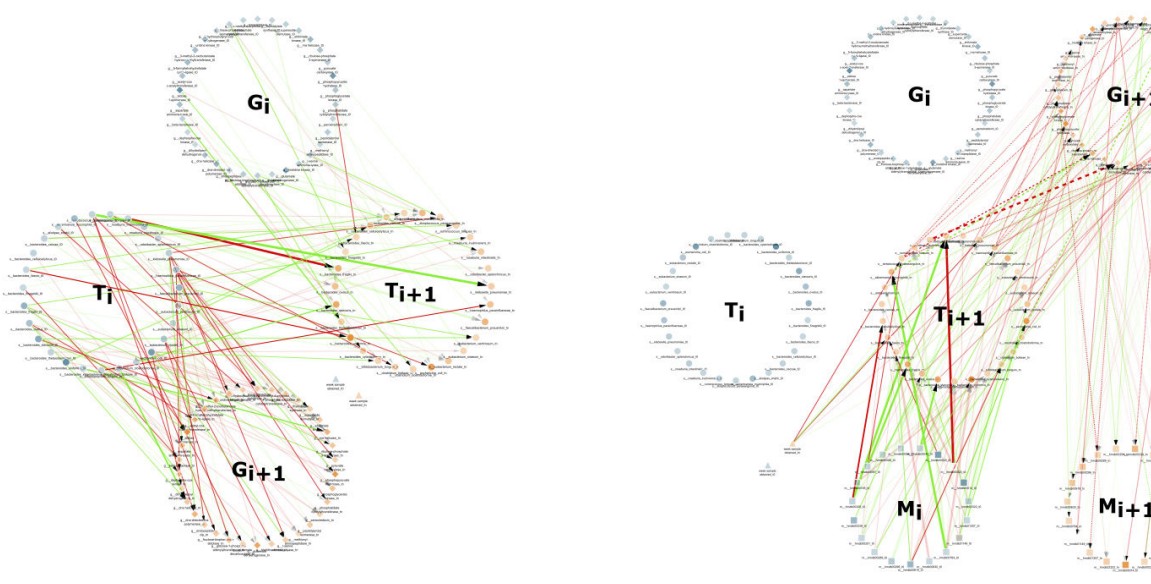

**FIG 1** Samples of the two-time-slice DBN networks for the four different multi-omic subsets produced by PALM. Self-edges are not displayed to avoid clutter. Networks were learned with a maximum number of parents of 3. The four networks show the nodes representing variables from each omics data source organized in two large circles, one representing the variables for the current time point (blue) and the other for the next time point (orange). Node shapes represent the omics data source of the variable. Taxa nodes are represented as filled circles, metabolites as filled squares, genes as filled diamonds, and clinical variables as filled triangles. Red (green) edges represent negative (positive resp.) regression coefficients. Edge width is proportional to the regression coefficient and edge opacity to the bootstrap score. Finally, node opacity is proportional to abundance. (a) DBN learned with just taxa abundance ($\mathbb{T}$). The data set included abundance of 27 bacteria and a clinical variable indicating the week the sample was obtained and resulted in a network with 95 edges. (b) DBN learned with taxa and metabolites ($\mathbb{TM}$). A set of 19 metabolites were added to the previous data set, and 164 edges were learned in this network. (c) DBN learned with the taxa and genes data set ($\mathbb{TG}$). A set of 34 genes were added to the taxa data set, and a network with 230 edges was learned. (d) DBN learned with the 27 taxa, 34 genes, and 19 metabolites ($\mathbb{TGM}$), resulting in a total of 311 edges.

remarkable information gain obtained by using additional omics data sets is readily observable in Fig. 1d), with a more complete picture of the state of the whole system, thus setting the stage for biologically relevant interpretations. The one non-omics variable (week of sample obtained), which is generically referred to as a "clinical variable," did not have any incident edges in the $\mathbb{TG}$ network, but it did in the other networks.

## Tool analysis

Network validation is a challenging problem because we do not have the ground truth network, which is what these methods try to approximate. In addition to analyzing the networks, the effect of the different network parameters was also explored. The heatmap in Fig. 2 shows the percentage of unrolling that is effected by METALICA on the networks learned by PyCausal (TETRAD). The columns labeled $\mathbb{TGMT}$, $\mathbb{TGT}$, and $\mathbb{TMT}$ represent the proportion of taxon to taxon interactions in the network learned with $\mathbb{T}$ that got unrolled with the networks learned with $\mathbb{TGM}$, $\mathbb{TG}$, and $\mathbb{TM}$, respectively. The alpha parameter for experiments with TETRAD is the significance threshold for the conditional independence tests.

The last column shows the average overall score of each unrolling, which is defined as the product of the individual bootstrap scores of the two replacement edges. Edge bootstrap scores represent the proportion of times an edge appears in bootstrap repetitions as described earlier.

Figure 3 shows the unrolling details output by METALICA in the experiments conducted with different methods, averaged over all parameters. All values except the last column represent the proportion of taxon to taxon interactions in the network learned with $\mathbb{T}$ that got unrolled with the networks learned with $\mathbb{TGM}$, $\mathbb{TG}$, and $\mathbb{TM}$, respectively. Tigramite networks showed the highest percentage of unrolled edges with $\mathbb{TGT}$ and $\mathbb{TMT}$ when compared with the other two methods, but fell short with $\mathbb{TGM}$, where DBNs resulted in a significantly higher percentage of unrolled edges. Note that applying temporal alignments to the data sets seemed to significantly improve the percentage of edges unrolled for the DBN method, especially with $\mathbb{TGMT}$, where the percentage rose from 24.7% to 78.8%. The increase was significantly lower with the other two datasets. The impact of temporal alignments on the other methods was inconsistent, where it showed both increase and decrease in the different columns. We also note that temporal alignments were used to normalize the "rates" of the underlying biological process of the different subjects.

## DISCUSSION

As shown in Fig. 2, as the alpha parameter decreases, the proportion of edges unrolled by METALICA decreases substantially. The smaller the alpha, the easier it is for two variables to be dependent, resulting in networks with more edges. This also means that higher alpha values result in networks with higher average confidence on each edge since it is also more difficult for it to be learned by chance. This is consistent with the higher percentage of unrolling for larger alpha values, indicating that the edges with

| Method | Temporal Alignment | Alpha | Proportion of unrolled edges | | | Overall Score |
|---|---|---|---|---|---|---|
| | | | $\mathbb{TGMT}$ | $\mathbb{TGT}$ | $\mathbb{TMT}$ | |
| PyCausal | No | 0.01 | 0.770 | 0.659 | 0.667 | 0.019 |
| PyCausal | No | 0.001 | 0.275 | 0.604 | 0.451 | 0.024 |
| PyCausal | No | 0.0001 | 0.058 | 0.391 | 0.333 | 0.060 |
| PyCausal | Yes | 0.01 | 0.724 | 0.711 | 0.158 | 0.039 |
| PyCausal | Yes | 0.001 | 0.288 | 0.750 | 0.231 | 0.055 |
| PyCausal | Yes | 0.0001 | 0.117 | 0.417 | 0.350 | 0.047 |

**FIG 2** Heatmap showing the proportion of edges unrolled by METALICA in the Crohn's disease data sets for the networks obtained from PyCausal (TETRAD) as the alpha parameter varies using data sets with and without temporal alignment. Last column shows the overall bootstrap score.

higher support get unrolled more frequently, adding support for the unrolling process. Interestingly, there is a clear reversal of the pattern for the overall bootstrap score (last column) for the experiments without temporal alignment, where, contrary to our intuition, the smaller alpha values result in higher overall scores. Interestingly, temporally aligning the data set seems to fix this problem, which would support the necessity of alignment as a pre-processing step.

Also, as shown in Fig. 3, the DBN/PALM method seems more stable than the other two algorithms, since the much higher average overall bootstrap score indicates that in each bootstrap, the edges learned are consistent with the ones learned in other bootstrap runs. This lower variability across the different random data subsamples used is a clear advantage of the DBN/PALM method.

The top unrollings and de-confoundings discovered by METALICA using the networks from all the methods were sorted based on the overall bootstrap score, and other factors like the number of networks they appear in, or the different network types that supported this particular finding. We discuss below some particularly interesting results from the METALICA analysis described above.

## Uncovering unrolled biological relationships

Here, we discuss the unrolling of specific edges from the METALICA results using the data set containing all diseases. First, we consider the edge *Eubacterium siraeum* → *Bacteroides thetaiotaomicron* in $G_T$, i.e., the edge between the abundance of the two bacterial taxa, *E. siraeum* and *B. thetaiotaomicron*. It manifests itself as the unrolled path *E. siraeum* → uridine kinase → cytidine → *B. thetaiotaomicron* in $G_{TGM}$, as shown in Fig. 4. The following is the support for each edge in the unrolled path from the literature and the knowledgebases. Both *E. siraeum* and *B. thetaiotaomicron* contain the gene to produce enzyme uridine kinase (54, 55). This enzyme, when present in prokaryotes and eukaryotes, phosphorylates both uridine and cytidine to their mono-phosphate forms, and vice versa. The specific reactions that this enzyme is capable of performing are the following (56–58):

 i. ATP + uridine ⇌ ADP +UMP and
 ii. ATP + cytidine ⇌ ADP + CMP,

where ATP stands for adenosine tri-phosphate, ADP stands for adenosine di-phosphate, UMP stands for uridine mono-phosphate, and CMP stands for cytidine

| Method | Temporal Alignment | Proportion of unrolled edges | | | Overall Score |
|---|---|---|---|---|---|
| | | $\mathbb{TGMT}$ | $\mathbb{TGT}$ | $\mathbb{TMT}$ | |
| PyCausal | No | 0.3675 | 0.5515 | 0.4835 | 0.0343 |
| PyCausal | Yes | 0.3763 | 0.6257 | 0.2462 | 0.0471 |
| Tigramite | No | 0.2000 | 0.7220 | 0.9449 | 0.0115 |
| Tigramite | Yes | 0.2000 | 0.6663 | 0.9186 | 0.0110 |
| DBN | No | 0.2472 | 0.3890 | 0.2136 | 0.4640 |
| DBN | Yes | 0.7879 | 0.4252 | 0.3343 | 0.3736 |

**FIG 3** Heatmap showing percentages of edges unrolled by METALICA in the Crohn's disease data sets for all the methods averaged over all parameter choices. The last column shows the overall bootstrap score.

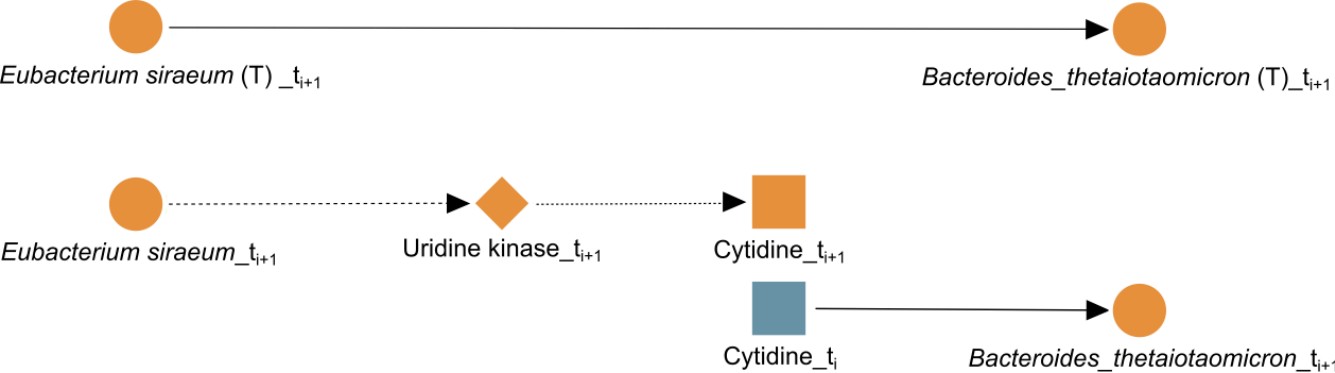

**FIG 4** Biologically confirmed unrolling. The edge *Eubacterium siraeum* → *Bacteroides thetaiotaomicron* learned in $G_{\mathbb{T}}$ (T) is unrolled into *Eubacterium siraeum* → uridine kinase → cytidine → *Bacteroides thetaiotaomicron* in $G_{\mathbb{TGM}}$.

mono-phosphate. Since *B. thetaiotaomicron* carries the gene for uridine kinase, it has the ability to perform the forward reaction and consume it by phosphorylating cytidine to CMP. More importantly, *B. thetaiotaomicron* also has the gene for cytidine deaminase, which scavenges exogenous and endogenous cytidine for UMP synthesis (59). The reaction performed by this enzyme is cytidine + $H_2O$ ⇌ uridine + ammonia (60–62), which validates the third and last edge (cytidine → *B. thetaiotaomicron*) in Fig. 4. In addition, experimental results show that a cytidine-scavenging system confers colonization fitness to *B. thetaiotaomicron* and, therefore, positively impact its abundance (63). Interestingly, uridine may be playing a role in this connection between the two taxa, since both enzymes discussed involve uridine, so both taxa can produce and consume uridine. Reinforcing this argument is the fact that the edge uridine → *B. thetaiotaomicron* is also present in the same network $G_{\mathbb{TGM}}$. Moreover, this unrolling can be important for IBD. Treatment for Crohn's disease with live *B. thetaiotaomicron* or its products displays strong efficacy in preclinical models of IBD, with multiple benefits (64). Similarly, there is precedent to treat gastrointestinal problems with *E. siraeum* (65), and activation-induced cytidine deaminase seems to prevent colon cancer development despite persistent inflammation in the colon (66).

In summary, our unrolling methods allow us to make biological sense out of a set of related edges in the series of networks generated from the multi-omics data.

As a second example, the path *Bacteroides stercoris* → uridine kinase → cytidine → *Bacteroides stercoris* can also be validated, which can be thought of as an unrolling of the self-loop from *Bacteroides stercoris* to itself in $G_{\mathbb{T}}$ as shown in Fig. 5. The taxon, *B. stercoris*, carries the gene for both uridine kinase (67) and cytidine deaminase (68), so it can both produce and consume cytidine, and since cytidine deaminase can scavenge endogenous cytidine, this lends further support to the self-loop edge from *B. stercoris* to itself; it might be regulating itself through the cytidine or uridine internally. Interestingly, *B. stercoris* is linked to colorectal cancer (69), and its increased abundance was detected in fecal samples of Crohn's disease (CD) patients (70). Also, an increased reactivity of immunoglobulin G from Crohn's disease patients toward *B. stercoris* and other species of *Bacteroides* has been shown in the serum of CD patients (71).

Two examples of "partial" validations of unrollings from our experiments are also provided. The unrolled path *Bacteroides finegoldii* → phosphatidate cytidylyltransferase → betaine → *Eubacterium ventriosum* was discovered by our search. It first appeared as an edge *B. finegoldii* → *E. ventriosum* in $\mathbb{T}$, which then got unrolled in $\mathbb{TG}$, $\mathbb{TM}$, and $\mathbb{TGM}$. *B. finegoldii* is an anaerobic gram-negative bacteria that has been found to be generally beneficial in the gut (72). It contains the gene BN532_01044 which expresses the phosphatidate cytidylyltransferase protein. This is a membrane-bound enzyme that participates in the glycerophospholipid metabolism and phosphatidylinositol signaling system. Moreover, *B. finegoldii* is known to produce the metabolite betaine (73).

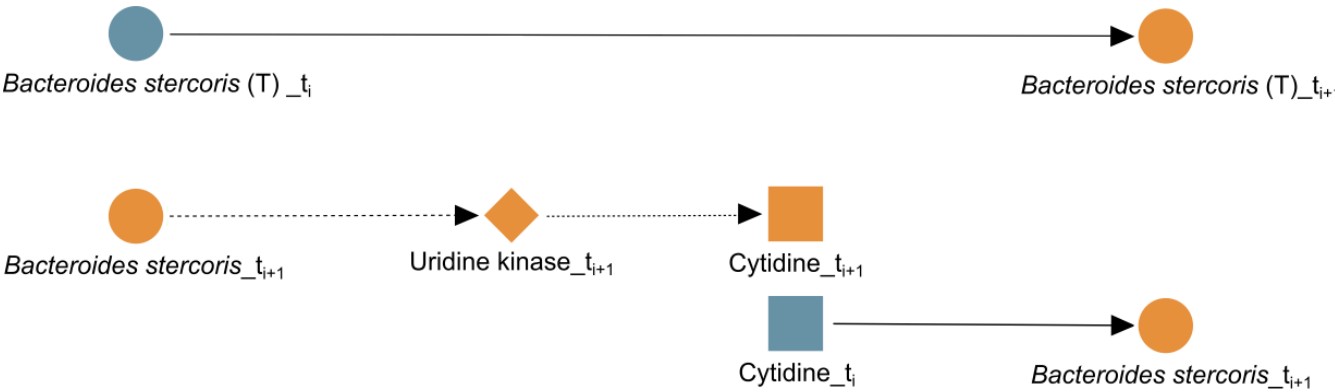

**FIG 5** Biologically confirmed unrolling. The edge *Bacteroides stercoris → Bacteroides stercoris* learned in $G_\mathbb{T}$ (T) is unrolled into *Bacteroides stercoris →* uridine kinase → cytidine → *Bacteroides stercoris* in $G_{\mathbb{TGM}}$

Increased levels of betaine have been found to benefit IBD patients, allowing for proper digestion and assimilation of nutrients. Over the last decade, doctors have recommended betaine-rich foods as a way to help IBD patients rapidly absorb and distribute vital vitamins and minerals needed to maintain diversity in the gut (73). Additionally, recent studies have shown betaine to be correlated to the *Eubacterium* genus and to be of general importance for the osmotic adaptation of most species of *Eubacterium* (74). Even though no specific study was found about the species *Eubacterium ventriosum*, the fact that betaine was found to increase the abundance of the *Eubacterium* genus lends support to the argument that *Eubacterium* members consume betaine through the conversion of acetate (75), thus partially validating the unrolling. Moreover, while acetate was not contemplated in the data set, one of its precursors, choline, was. Many strong unrollings have a link from choline to a member of the *Eubacterium* genus in the data set (*E. ventriosum, E. siraeum, E. rectale*), and almost every method learned the edge betaine → *E. ventriosum* as part of specific unrollings, which could be an indication of a pathway transforming choline to acetate to betaine, which may be facilitated by members of the genus, *Eubacterium*.

The path *Bacteroides ovatus* → DNA helicase → pyridoxine → *Bacteroides ovatus* in $\mathbb{TGM}$ can be thought of as an unrolling of a self-loop edge in $\mathbb{T}$ from *B. ovatus* to itself, which got unrolled in $\mathbb{TG}, \mathbb{TM}$, and $\mathbb{TGM}$. Moreover, *B. ovatus* is present in the gut microbiome and plays a crucial role in the dysbiosis of the gut health. This anaerobic bacteria has been found to have significantly elevated abundance in patients suffering from IBD. Findings suggest that some species of *Bacteroides* injure gut tissue and induce inflammation (76). This bacterium does carry the gene *dnaB*, which expresses the protein DNA helicase, an enzyme responsible in unpacking genes in an organism and DNA repair. The production of the metabolite pyridoxine has been found in great proportion when there is an abundance of *B. ovatus* (77). However, evidence suggesting the consumption of pyridoxine by the taxa could not be found. When pyriodoxine is present in great abundance, it is involved in many biochemical pathways that lead to the synthesis or metabolism of nucleic acids, immune modulatory metabolites, and many others (77). However, when scarce, it leads to inflammation. We consider this as another example of a "partial" validation of our unrolling strategy.

### Uncovering de-confounded biological relationships

We focus next on the de-confounding actions performed by METALICA on the networks obtained using the data set containing all diseases. The edge thymidylate synthase → glutamate dehydrogenase was inferred in the $\mathbb{G}$ network but disappeared in the $\mathbb{TG}$ network, possibly because both genes are present in the taxon *Haemophilus parainfluenzae*. This suggests that the suggested relationship between the two genes is spurious

and the taxon is the confounder. *H. parainfluenza* is an opportunistic pathogen that has been found in elevated levels in patients suffering from many diseases including pneumonia and conjunctivitis. Recent studies have shown that a high abundance of this pathogen was found in patients suffering from IBD. Different dynamics have been noted for the abundance of *H. parainfluenza* in the literature. For instance, when IBD patients enter remission, there is a steep decline in this pathogen (78). Additionally, the two genes that are present in *H. parainfluenzae* were found to produce proteins that help drive diseases including colon cancer.

## Limitations and future work

The methods used by METALICA are only applicable to multi-omic data sets, which are relatively uncommon. However, this is expected to change in the near future with the increased effort to understand the underlying mechanisms within biological processes. Second, these methods do not provide definitive evidence for the causal chains, but rather lend support to generate hypotheses that would have to be proved with experiments in the laboratory. We argue that as larger data sets become more and more commonplace, METALICA will become increasingly useful.

## Conclusion

We have developed METALICA, which consists of two novel *post hoc* network analysis algorithms, namely, unrolling and de-confounding. We first learned biological networks from a longitudinal multi-omic IBD data set with three state-of-the-art network and causal discovery tools. We then applied METALICA to the networks learned by the tools (DBN/PALM, tsGFCI/TETRAD, and Tigramite) and compared their predictive performance. The networks produced using DBN/PALM produced the most number of unrollings, suggesting that even though the tool was not explicitly built for causal discovery, its conditional probability underpinnings produce edges that have a reasonable chance of representing causal relationships and lead to further biological discoveries as outlined above. The top findings by our algorithms were analyzed, and relevant biological interpretations were presented for specific network-inferred interactions.

### ACKNOWLEDGMENTS

This work was partially supported by NIH 1R15AI128714-01 (G.N.) and the FIU Dissertation Year Fellowship (D.R.-P.). The funders had no role in study design, data collection and interpretation, or the decision to submit the work for publication.

### AUTHOR AFFILIATIONS

[1]Bioinformatics Research Group (BioRG), Florida International University, Miami, Florida, USA
[2]Florida International University, Miami, Florida, USA
[3]Biomolecular Sciences Institute, Florida International University, Miami, Florida, USA

### AUTHOR ORCIDs

Daniel Ruiz-Perez ⓘ https://orcid.org/0000-0002-5622-560X
Kalai Mathee ⓘ http://orcid.org/0000-0003-4569-5419
Giri Narasimhan ⓘ http://orcid.org/0000-0003-0535-4871

### FUNDING

| Funder | Grant(s) | Author(s) |
| --- | --- | --- |
| HHS \| NIH \| OSC \| Common Fund (NIH Common Fund) | 1R15AI128714-01 | Giri Narasimhan |

| Funder | Grant(s) | Author(s) |
|---|---|---|
| University Graduate School, Florida International University (UGS) | Dissertation Year Fellowship | Daniel Ruiz-Perez |

## AUTHOR CONTRIBUTIONS

Daniel Ruiz-Perez, Conceptualization, Data curation, formal analysis, investigation, methodology, Project administration, Resources, software, Validation, visualization, Writing – original draft, Writing – review and editing | Isabella Gimon, Validation | Musfiqur Sazal, Conceptualization, Data curation | Kalai Mathee, Supervision | Giri Narasimhan, Conceptualization, Funding acquisition, Project administration, Resources, Supervision, Writing – original draft, Writing – review and editing

## DATA AVAILABILITY

All code, networks, and longitudinal microbiome data sets are available from https://github.com/DaniRuizPerez/Metalica_Public. All data analyzed in this work were derived from the iHMP IBD website: https://www.ibdmdb.org (18).

## ADDITIONAL FILES

The following material is available online.

### Supplemental Material

**Supplemental material (mSystems01303-23-s0001.pdf).** Organization of Metalica website, details on experiments, and settings of all parameters in experiments.

### Open Peer Review

**PEER REVIEW HISTORY (review-history.pdf).** An accounting of the reviewer comments and feedback.

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
