## [Reviewer comments · mSystems]

Unfolding and De-confounding: Biologically meaningful causal inference from longitudinal multi-omic networks using METALICA

Daniel Ruiz-Perez, Isabella Gimon, Musfiqur Sazal, Kalai Mathee, and Giri Narasimhan

Corresponding Author(s): Giri Narasimhan, Florida International University

Review Timeline:

Submission Date:	December 19, 2023
Editorial Decision:	February 20, 2024
Revision Received:	June 13, 2024
Accepted:	July 10, 2024

Editor: Hulin Li

Reviewer(s): Disclosure of reviewer identity is with reference to reviewer comments included in decision letter(s). The following individuals involved in review of your submission have agreed to reveal their identity: Sanguthevar Rajasekaran (Reviewer #2)

Transaction Report:

DOI: <https://doi.org/10.1128/msystems.01303-23>

Re: mSystems01303-23 (Unfolding and De-confounding: Biologically meaningful causal inference from longitudinal multi-omic networks using METALICA)

Dear Dr. Giri Narasimhan:

Thank you for the privilege of reviewing your work. Below you will find instructions from the mSystems editorial office, and the reviewer comments.

Revision Guidelines

Sincerely,
Huilin Li
Editor
mSystems

Reviewer #1 (Comments for the Author):

General comments:

This paper presents two techniques, unrolling and de-confounding, for uncovering biological entities considered to be confounders when using standard algorithms for causal inference. The authors test the techniques on a publicly available longitudinal multiomics microbiome dataset. Finally, the paper is clear and easy to read.

Major comments:

- The main technical contribution is rather limited.
- While I understand the rationale of the two techniques, the application of the techniques is very ad-hoc. It also lacks any background distribution to assess significance. Additionally, it is unclear how to handle situations where there are multiple decompositions for the same taxa relationships.
- I would suggest adding a section on synthetic data that demonstrates the behavior of the techniques across different biologically-relevant scenarios.
- Another issue is the granularity of the sampling rate. Just because you can explain a taxa relationship through a metabolite, it does not guarantee that there aren't other causal relationships at a sampling rate.

Reviewer #2 (Comments for the Author):

A major goal of this study is to infer biologically meaningful multi-omic interactions. Specifically, the authors want to discover interactions among microbial taxa, expressed genes, metabolites consumed and/or produced, etc. Toward this goal, they have created a software package called METALICA. METALICA employs novel unrolling and de-confounding techniques used to uncover multi-omic entities that are believed to act as confounders for some of the relationships that may be inferred using standard causal inferencing tools. Existing literature seems to mostly focus on the analysis of each omic data separately. Thus the proposed approach is warranted.

The authors have tested METALICA on multi-omic data sets resulting from an iHMP longitudinal study of Inflammatory Bowel Disease (IBD) microbiomes. They have verified the most significant unrollings and de-confoundings using the literature and public databases.

The main idea of unrolling is to infer whether and how two taxa interact with each other using (multiple) networks learnt using different subsets of the omic data. For example, let G_1 and G_2 be two different networks learnt (using different subsets of data). If (a, b) is an edge in G_1 , this edge does not occur in G_2 , and (a, c) and (c, b) are two edges present in G_2 , then we can infer that a and b interact via c . De-confounding is similar to unrolling (in that information from different networks are fused to make new inferences). The authors employ three network learning methods, namely, Dynamic Bayesian Networks, TETRAD, and Tigramite.

In summary, the methods proposed in the paper are novel and timely, given that existing approaches for the integrated analysis of multiple omic datasets are sparse. Experimental results indicate the effectiveness of METALICA. As a result, this referee recommends acceptance of this paper for mSystems.

Review on

Unfolding and De-confounding: Biologically meaningful causal inference from longitudinal multi-omic networks using METALICA

by D. Ruiz-Perez , I. Gimon , M. Sazal , K. Mathee, and G. Narasimhan (corr-auth)

A major goal of this study is to infer biologically meaningful multi-omic interactions. Specifically, the authors want to discover interactions among microbial taxa, expressed genes, metabolites consumed and/or produced, etc. Toward this goal, they have created a software package called METALICA. METALICA employs novel unrolling and de-confounding techniques used to uncover multi-omic entities that are believed to act as confounders for some of the relationships that may be inferred using standard causal inferencing tools. Existing literature seems to mostly focus on the analysis of each omic data separately. Thus the proposed approach is warranted.

The authors have tested METALICA on multi-omic data sets resulting from an iHMP longitudinal study of Inflammatory Bowel Disease (IBD) microbiomes. They have verified the most significant unrollings and de-confoundings using the literature and public databases.

The main idea of unrolling is to infer whether and how two taxa interact with each other using (multiple) networks learnt using different subsets of the omic data. For example, let G_1 and G_2 be two different networks learnt (using different subsets of data). If (a, b) is an edge in G_1 , this edge does not occur in G_2 , and (a, c) and (c, b) are two edges present in G_2 , then we can infer that a and b interact via c . De-confounding is similar to unrolling (in that information from different networks are fused to make new inferences). The authors employ three network learning methods, namely, Dynamic Bayesian Networks, TETRAD, and Tigramite.

In summary, the methods proposed in the paper are novel and timely, given that existing approaches for the integrated analysis of multiple omic datasets are sparse. Experimental results indicate the effectiveness of METALICA. As a result, this referee recommends acceptance of this paper for mSystems.

Reviews

We are very appreciative of the comments from the reviewers. We have done our best to respond to them below.

Reviewer #1 (Comments for the Author):

General comments:

This paper presents two techniques, unrolling and de-confounding, for uncovering biological entities considered to be confounders when using standard algorithms for causal inference. The authors test the techniques on a publicly available longitudinal multiomics microbiome dataset. Finally, the paper is clear and easy to read.

Major comments:

- The main technical contribution is rather limited.

We note that while the technical contributions may seem limited, its strength lies in its simplicity and its general applicability to many situations.

- While I understand the rationale of the two techniques, the application of the techniques is very ad-hoc. It also lacks any background distribution to assess significance. Additionally, it is unclear how to handle situations where there are multiple decompositions for the same taxa relationships.

- I would suggest adding a section on synthetic data to demonstrate the behavior on the techniques across different biologically-relevant scenarios.

This is a reasonable request in principle. We spent a lot of time looking for and experimenting with umpteen tools to design a meaningful “synthetic” experiment to validate our work. Our final conclusion from our efforts is as follows.

Our tool, METALICA, sits on top of other tools (Palm, TETRAD, Tigramite) that have already been validated, some even with synthetic data. We are not aware of any reasonable tools that can generate synthetic longitudinal microbiome data sets in a biologically-relevant manner. We were able to locate tools that generate random samples from a (random) Bayesian network, but none that generate a random time series. Even more critical is that even if one were to be located, there are no sample generation tools that would challenge the unrolling scenario. In other words, if there is a chain $A \rightarrow B \rightarrow C$ without the edge $A \rightarrow C$, and if $A \rightarrow C$ is always produced if B is suppressed in the chain, then the algorithm in METALICA will always (deterministically) determine that B is an intermediate and therefore will not be challenged. If the edge $A \rightarrow C$ is not always produced, then it is merely testing the causality tool or the sample generation technique, not our inference tool. Similar arguments can be made for the deconfounding scenario, making the synthetic data test

superfluous. Basically, in this work we are not aiming at inferring the network causal structure underlying a data set, but at augmenting the different networks generated by already published and tested causal network inference methods. The algorithms we propose take as input already learned causal networks, so the process of learning those networks is not entirely relevant to this work.

By the same token, the statistical significance of any inference is merely dependent on that of the causal discovery tools we use.

- Another issue is the granularity of the sampling rate. Just because you can explain a taxa relationship through a metabolite, it does not guarantee that there aren't other causal relationships at a sampling rate.

Once again, we do not see this as a limitation of the tool METALICA, but of the underlying causality tools themselves. For Tigramite and Tetrad, it can only discover causal relationships for the data that is provided to it. If data with other sampling rates are available then it is possible that better inference is possible. The DBN-based tool, Palm, can internally modify the sampling rate because it takes time series data and converts them to continuous functions after smoothing and temporal alignments. However, this too is not relevant to the working of METALICA (since it builds on top of Palm, which was already published), which tries to validate the causal network structure, not extract it from data.

Reviewer #2 (Comments for the Author):

A major goal of this study is to infer biologically meaningful multi-omic interactions. Specifically, the authors want to discover interactions among microbial taxa, expressed genes, metabolites consumed and/or produced, etc. Toward this goal, they have created a software package called METALICA. METALICA employs novel unrolling and de-confounding techniques used to uncover multi-omic entities that are believed to act as confounders for some of the relationships that may be inferred using standard causal inferencing tools. Existing literature seems to mostly focus on the analysis of each omic data separately. Thus the proposed approach is warranted.

The authors have tested METALICA on multi-omic data sets resulting from an iHMP longitudinal study of Inflammatory Bowel Disease (IBD) microbiomes. They have verified the most significant unrollings and de-confoundings using the literature and public databases.

The main idea of unrolling is to infer whether and how two taxa interact with each other using (multiple) networks learnt using different subsets of the omic data. For example, let G_1 and G_2 be two different networks learnt (using different subsets of data). If (a, b) is an edge in G_1 , this edge does not occur in G_2 , and (a, c) and (c, b) are two edges present in G_2 , then we can infer that a and b interact via c . De-confounding is similar to unrolling (in that

information from different networks are fused to make new inferences). The authors employ three network learning methods, namely, Dynamic Bayesian Networks, TETRAD, and Tigramite.

In summary, the methods proposed in the paper are novel and timely, given that existing approaches for the integrated analysis of multiple omic datasets are sparse. Experimental results indicate the effectiveness of METALICA. As a result, this referee recommends acceptance of this paper for mSystems.

We appreciate the comments from the reviewer and for capturing the core ideas from the manuscript. We agree that the main idea is to build simple, yet powerful, inferencing from what we learn from subsets of omics data.

Re: mSystems01303-23R1 (Unfolding and De-confounding: Biologically meaningful causal inference from longitudinal multi-omic networks using METALICA)

Dear Dr. Giri Narasimhan:

Your manuscript has been accepted, and I am forwarding it to the ASM production staff for publication. Your paper will first be checked to make sure all elements meet the technical requirements. ASM staff will contact you if anything needs to be revised before copyediting and production can begin. Otherwise, you will be notified when your proofs are ready to be viewed.

Sincerely,
Huilin Li
Editor
mSystems

Reviewer #1 (Comments for the Author):

I still think the author should provide some guidance on when the method is reliable.

Reviewer #2 (Comments for the Author):

The authors have responded adequately to the comments.